Diversity of compounds in femoral secretions of Galápagos iguanas (genera: Amblyrhynchus and Conolophus), and their potential role in sexual communication in lek-mating marine iguanas (Amblyrhynchus cristatus)

Ibáñez Alejandro 1 a.ibanez-ricoma@tu-braunschweig.de
Menke Markus 2
Quezada Galo 3
Jiménez-Uzcátegui Gustavo 4
http://orcid.org/0000-0002-4810-324X Schulz Stefan 2
Steinfartz Sebastian 1
1 Division of Evolutionary Biology, Zoological Institute, Technische Universität Braunschweig , Braunschweig , Germany
2 Institute of Organic Chemistry, Technische Universität Braunschweig , Braunschweig , Germany
3 Dirección Parque Nacional Galápagos , Puerto Ayora, Santa Cruz, Galápagos , Ecuador
4 Charles Darwin Research Station , Puerto Ayora, Galápagos , Ecuador
Measey John
Electronic publication date: 2017 Aug 17
Publication date: 2017
Volume: 5
Electronic Location ID: e3689
Received 2017 Jun 12; Accepted 2017 Jul 24
Copyright: © 2017 Ibáñez et al.
Copyright year: 2017
Copyright holder: Ibáñez et al.
License: This is an open access article distributed under the terms of the Creative Commons Attribution License, which permits unrestricted use, distribution, reproduction and adaptation in any medium and for any purpose provided that it is properly attributed. For attribution, the original author(s), title, publication source (PeerJ) and either DOI or URL of the article must be cited.
License URL: https://creativecommons.org/licenses/by/4.0/

Keywords: Femoral glands, Lek-mating system, Lipophilic fraction, Body condition index, 11-Eicosenoic acid, Genetic population structure

Funding: Swiss Friends of the Galápagos Islands Alexander von Humboldt Foundation This work was supported by a grant from the Swiss Friends of the Galápagos Islands and by a postdoctoral research fellowship of the Alexander von Humboldt Foundation to AI. The funders had no role in study design, data collection and analysis, decision to publish, or preparation of the manuscript.

==============================
Background

Chemical signals are widely used in the animal kingdom, enabling communication in various social contexts, including mate selection and the establishment of dominance. Femoral glands, which produce and release waxy secretions into the environment, are organs of central importance in lizard chemical communication. The Galápagos marine iguana (Amblyrhynchus cristatus) is a squamate reptile with a lek-mating system. Although the lekking behaviour of marine iguanas has been well-studied, their potential for sexual communication via chemical cues has not yet been investigated. Here we describe the diversity of the lipophilic fraction of males’ femoral gland secretions among 11 island populations of marine iguanas, and compare it with the composition of its sister species, the Galápagos land iguana (Conolophus subcristatus). We also conducted behavioural observations in marine iguana territorial males in order to explore the possible function of these substances in the context of male dominance in leks.

Methods

Femoral secretions were analysed by gas chromatography coupled to mass spectrometry (GC–MS), and chromatography with a flame ionisation detector (GC-FID) in order to characterise the lipophilic composition. To understand the potential role of femoral secretions in marine iguana intraspecific communication, territorial males were sampled for their femoral glands and monitored to record their head bob rate—a territorial display behaviour in males—as well as the number of females present in their leks.

Results

We found that the gland secretions were composed of ten saturated and unsaturated carboxylic acids ranging in chain length between C16 and C24, as well as three sterols. Cholesterol was the main compound found. Intriguingly, land iguanas have a higher diversity of lipophilic compounds, with structural group of lipids (i.e. aldehydes) entirely absent in marine iguanas; overall the chemical signals of both species were strongly differentiated. Lipid profiles also differed among populations of marine iguanas from different islands, with some islands demonstrating a high diversity of lipophilic compounds (i.e. full spectra of compounds), and others lacking one or more compounds. Among the compounds most frequently found missing were 11- and 13-eicosenoic acids. Gland secretions of males with a better body condition and with a higher dominance status (i.e. those accompanied by females and with higher head bob display) were proportionately richer in C20-unsaturated fatty acids (11-eicosenoic acid).

Discussion

Land and marine iguanas strongly diverged in their chemical composition of the femoral glands likely due to ecological differences between both species. Despite that marine iguana populations varied in their femoral gland composition that was not related to their genetic structure. Our results indicated that 11-eicosenoic acid may play an important role in intraspecific chemical communication in marine iguanas.

Introduction

Intraspecific chemical communication is a widespread phenomenon among vertebrates (Wyatt, 2003). Although reptiles rely heavily on visual signals to communicate with conspecifics, they also use chemical signals in sexual interactions, as well as in a broader social context (Mason & Parker, 2010). Squamate reptiles (lizards and snakes) are suitable models for researchers aiming to study chemical communication. This is mainly due to highly developed chemosensory capacity, which is used for intraspecific communication. For instance, substrate-deposited scents are used by snakes to locate partners and/or avoid rivals (LeMaster & Mason, 2003; Shine et al., 2005). Lizards have the ability to produce scent marks that are recognised by conspecifics (Carazo, Font & Desfilis, 2008; Khannoon, El-Gendy & Hardege, 2011; Martins et al., 2006). The popularity of lizards as models to study chemical communication has been increasing steadily. However, the functions of the specific compounds that make up chemical signals remain poorly known.

One of the most important organs involved in chemical communication are femoral glands, which are epidermal skin glands situated ventrally on the hind limbs (Mayerl, Baeckens & Van Damme, 2015). Reptiles release waxy secretions from their femoral glands that are composed of a mixture of proteins and lipids (Weldon, Flachsbarth & Schulz, 2008). For example, desert iguanas (Dipsosaurus dorsalis) have heavy femoral secretions, composed of a 20% lipid fraction and 80% of protein material (Alberts, 1990). Proteins absorb ultraviolet light that could be used by desert iguanas to locate pheromones of low volatility (Alberts, 1989). Inter-species variability in proteins from femoral glands suggests that proteins could be used for species discrimination in sympatrically occurring lizard species (Alberts, 1991). A recent study on Turks and Caicos iguanas (Cyclura carinata) showed that genetic differentiation among lineages from different islands is reflected by the pattern of proteins in femoral secretions (Welch et al., 2017). In addition, the lipophilic compounds in the femoral secretions play an important role in many types of sexually mediated behaviour. For example, femoral glands in males of some lizard species emit chemicals that could communicate specific information about the traits of the signaller, and may therefore guide female mate choices (López & Martín, 2005; Martín & López, 2006a, 2011; Olsson et al., 2003). A link between chemical signals and dominance has also been suggested. For instance, femoral gland secretions of male Iberolacerta monticola convey information about the individual’s dominance status (Moreira, López & Martín, 2006). This situation is hardly surprising, given that the function and activity of femoral glands are regulated by androgens (Fergusson, Bradshaw & Cannon, 1985). In green iguanas, Iguana iguana, the level of testosterone is positively correlated with the amount of lipids in the femoral secretions of dominant males (Alberts, Pratt & Phillips, 1992). In this framework, several studies have described the lipid components of the femoral glands in various species of lizards (reviewed in Martín & López (2014) and Weldon, Flachsbarth & Schulz (2008)). Although most research on the specific chemicals that compose femoral secretions has focused on lacertids (Heathcote et al., 2014; Martín et al., 2015; García-Roa et al., 2017), a few older studies about iguanids are also available (Alberts, 1990; Alberts, Pratt & Phillips, 1992; Weldon et al., 1990). The femoral secretions of the green iguana I. iguana are known to contain a high diversity of volatile lipids that appear to be received by chemoreception (Alberts, 1993; Alberts, Pratt & Phillips, 1992). These lipids include several saturated and unsaturated fatty acids of different chain lengths (C14–C26), as well as sterols (Alberts, Pratt & Phillips, 1992; Weldon et al., 1990).

Despite increasing interest in the chemical ecology of lizards, there are no such studies on one of the most outstanding reptile species in the world: the marine iguana (Amblyrhynchus cristatus). This species is endemic to the Galápagos Islands—a world-renowned biodiversity hotspot formed by 13 major islands (>10 km2), six smaller ones and numerous islets in the eastern Pacific. One of the most interesting aspects of the species’ behavioural ecology is its lek-mating system. A lek is defined as a mating system in which there is no parental care, males aggregate in clusters, and females benefit from no resources other than the gametes available in the male territories (Höglund & Alatalo, 1995). Lek-mating behaviour has not been studied in detail for most of reptiles. In contrast, marine iguana lekking has received considerable attention over the past decades, and the system is reasonably well described. At the beginning of the mating season, male iguanas establish small clustered territories along the rocky coastline. These territorial males display head-bobbing behaviour in order to attract females to their territories, presumably to mate and also to intimidate competing males that are approaching the territory of the dominant male (Wikelski, Carbone & Trillmich, 1996). However, males that cannot establish territories (marginal males) have far less reproductive success than territorial ones, and basically rely on forced copulations with females that remain outside of the lek territories (Partecke, von Haeseler & Wikelski, 2002; Wikelski, Carbone & Trillmich, 1996). Apparently, females may benefit from mating within specific males’ territories, as the territorial male protects them from harassment by others (Trillmich & Trillmich, 1984; Wikelski, Carbone & Trillmich, 1996). The importance of chemical signals in lekking species has been verified in insects. The mosquito Aedes aegypti uses an aggregation-pheromone to mediate lek establishment (Cabrera & Jaffe, 2007) and males of the sand fly Lutzomyia longipalpis may use pheromones to attract females to their leks (Jones & Hamilton, 1998). Therefore, the lek-mating system offers a valuable natural setup to investigate questions regarding chemical signalling.

Several factors suggest that marine iguanas might rely heavily on pheromones for intraspecific communication. Firstly, they possess femoral glands that are the source of chemical signals for many species of lizards (Mayerl, Baeckens & Van Damme, 2015). Secondly, observations have revealed a “substrate-licking” behaviour in marine iguanas (Carpenter, 1966) observed that male iguanas protrude their tongue onto the lava rocks as if testing for scent marks. Further, female iguanas have been reported to lick the sand intensively when exploring new areas to lay eggs (Rauch, 1988). Tongue licking is a chemoreceptive response to recognition of marked substrates in iguanids (De Fazio et al., 1977; Krekorian, 1989) and suggests that Amblyrhynchus cristatus might use conspecific olfactory cues for intraspecific communication.

Our study is the first to analyse the lipophilic fraction of femoral gland secretions in Galápagos iguanas. By performing intensive sampling of all major island populations and subspecies of marine iguanas (see Miralles et al., 2017), we seek to characterise the lipophilic profile of femoral glands, as well as to explore the variation between populations originating from distinct islands of the archipelago. In addition to sampling marine iguanas, we also collected femoral secretions of land iguanas (Conolophus subcristatus). Conolophus is the sister lineage of Amblyrhynchus, having diverged around 4.52 million years ago, most likely on the Galápagos Archipelago (MacLeod et al., 2015). Therefore, we compare the chemical profile of marine iguanas with their closest taxon, to investigate the level of differentiation in chemical signals between the two. Marine iguanas have a great mobility and could even be passively dispersed by oceanic currents (Carpenter, 1966; Higgins, 1978; Lanterbecq et al., 2010). Therefore, we expect that chemical composition has a low variation among populations of marine iguanas. In contrast, we hypothesise that land and marine iguanas diverge strongly in their chemical signals because they occupy distinct habitats and there is no gene flow among both species.

A further aim of this study was to understand the functions of femoral gland compounds in marine iguanas. For this purpose, we carried out behavioural observations in one population of Amblyrhynchus cristatus. Morphological and behavioural data of territorial males were collected during the breeding season and correlated with data on gland chemistry. Body size, as well as the body condition index, of male marine iguanas were estimated as determinant factors for female mate choice and male dominance hierarchy (Wikelski, Carbone & Trillmich, 1996). Territorial males were regularly monitored to estimate their head-bobbing rate, and the presence of females in their territories was recorded. Head-bobbing is a behaviour typically displayed by territorial males to defend their territories from other males, as well as to attract females during the breeding season (Wikelski, Carbone & Trillmich, 1996). Dominant males, i.e. those with high display levels (i.e. frequent head bobbing) and who are accompanied by females, should release certain chemicals in their territories that inform conspecifics of their status.

Materials and Methods

Femoral gland sampling

During the months of December and January of the years 2014–2016 (a period coinciding with the mating season of marine iguanas), an overall of 196 iguanas were captured in 11 islands of the Galápagos archipelago in order to sample their femoral glands. Of those, a subsample of 134, representing all major island populations and subspecies (see Table S1 and Fig. 1 for an overview), were used to examine the variation of chemical profiles among populations. However, due to uneven sampling effort, the number of obtained samples per population was unequal. Therefore, a subset of 103 samples, with even sample size for each population was randomly selected and used in statistical analyses in order to avoid any effects of an unbalanced design. The remaining samples (62) were taken from territorial males of a single population (i.e. Amblyrhynchus cristatus mertensi from La Lobería on San Cristóbal) and used to understand the function of chemical signals in intraspecific communication. In this way, chemical composition of femoral secretions could be analysed in relation to individual morphology and behaviour (see below: Morphological dataset and Behavioural dataset). The marine iguanas were captured using a pole with a lasso, and their femoral secretions were obtained by gently squeezing the femoral glands. Femoral secretions were then placed in a 2 mL glass vial with dichloromethane. All samples were stored cold (in portable coolers at approximately 4–8 °C) during the field work, and subsequently transported to laboratories in Germany where they were stored at −80 °C, until chemical analysis was performed.

Figure 1 Map of the Galápagos archipelago, showing the total number of femoral gland samples of A. cristatus collected per island, and the number of samples considered for statistical analysis in brackets.

Island abbreviations: GEN, Genovesa; MAR, Marchena; PIN, Pinta; SAN, Santiago; CRUZ, Santa Cruz; SRL, San Cristóbal (La Lobería); SRPC, San Cristóbal (Punta Pitt); SFE, Santa Fe; ESP, Española; FL, Floreana; IS, Isabela; FDA, Fernandina.

In addition, during January 2016, we also sampled the femoral glands of three individuals of land iguana Conolophus subcristatus, which are housed in a spacious outdoor enclosure at the Charles Darwin Research Station on Santa Cruz Island.

The Galápagos National Park authority granted the research permission for this study (permit numbers: PC-22-14, PC-08-15 and PC-09-16).

Chemical analysis and lipid characterisation

The chemical analyses were performed by gas chromatography coupled with mass spectrometry (GC–MS), and gas chromatograph with a flame ionisation detector (GC-FID). GC–MS analyses of natural compounds were performed on a GC 7890A/MSD 5975C from Agilent. Mass spectrometry was performed in an electron ionisation mode (EI) with 70 eV. Fused-silica capillary columns and HP-5MS (30 m, 0.25 mm i.D. 0.25 μm film thickness; Agilent Technologies, Santa Clara, CA, USA) were used, with helium as the carrier gas. GC-FID was performed using a HP-5 column (30 m, 0.25 mm i.D. 0.25 μm film thickness; Agilent Technologies, Santa Clara, CA, USA) using hydrogen (H2) as the carrier gas with a flame ionisation detection (FID) system.

The temperature program for the GC–MS was: 50-5-5-320 (50 °C starting temp., hold time 5 min then 5 °C/min increase up to 320 °C). For GC-FID with autosampler the temperature program 125-5-5-320 (125 °C starting temp., hold time 5 min, the 5 °C/min increase up to 320 °C) was used.

The samples collected in December 2014 and January 2015 were preserved in 500 μL dichloromethane. Aliquots of 20 μL from each sample were taken for derivatizations, and 0.8 μL of internal standard (tridecyl acetate) was added. The samples collected in December 2015 and January 2016 were stored in 200 μL dichloromethane. Aliquots of 5 μL were taken for derivatizations and 0.4 μL of internal standard (tridecyl acetate) was added, adding extra 20 μL dichloromethane. The addition of the internal standard ensured that similar concentrations of the samples were injected into the GC/MS system.

Derivatizations of the samples were performed using diazomethane (CH2N2), synthesised as an etheric solution from Diazald following Black’s procedure for converting carboxylic acids into their corresponding methyl esters (Black, 1983). Elucidation of the structures was performed by mass spectral comparison with data bases, and comparison with reference compounds identified 13 major compounds. Fatty acids were identified as their corresponding methyl esters (Table 1). In contrast to marine iguanas, land iguana samples also feature aldehydes in addition to carboxylic acids (Table 1). Besides the additional aldehydes, hexadecyl hexadecanoate and di-(9-octadecenoyl)-glycerol were also found (Table 1). The position of the double bonds was confirmed by dimethyl disulphide (DMDS)/I2-derivatization of methylated natural samples according to published procedures (Bruns et al., 2013; Buser et al., 1983).

Table 1 Percentage (mean ± SD) of major compounds present in Amblyrhynchus cristatus and Conolophus subcristatus secretions, with the retention indices (RI) of methyl esters of the fatty acids and cholesterol derivatives.

Lipids (mean ± SD)	Compound name	RI	
Marine iguana	Land iguana			
–	2.59 ± 1.65	Hexadecanal*	1,813	
–	0.55 ± 0.74	7-Hexadecenoic acid	1,904	
–	1.48 ± 1.08	9-Hexadecenoic acid	1,908	
22.41 ± 12.01	19.99 ± 12.19	Hexadecanoic acid	1,928	
–	0.48 ± 0.83	9-Octadecenal*	1,993	
–	0.27 ± 0.47	11-Octadecenal*	1,997	
–	0.59 ± 0.67	Octadecanal*	2,016	
–	1.55 ± 0.37	Di-(9-Octadecenoyl)-glycerol	2,473	
13.68 ± 4.81	20.40 ± 12.34	9-Octadecenoic acid	2,099	
1.45 ± 1.25	4.72 ± 2.80	11-Octadecenoic acid	2,104	
12.12 ± 7.15	8.37 ± 4.21	Octadecanoic acid	2,125	
4.85 ± 2.81	1.92 ± 1.06	5,8,11,14-Eicosatetraenoic acid	2,258	
0.35 ± 1.01	0.26 ± 0.15	11-Eicosenoic acid	2,299	
0.1 ± 0.23	–	13-Eicosenoic acid	2,306	
3.43 ± 2.58	0.55 ± 0.31	Eicosanoic acid	2,326	
2.41 ± 1.75	0.47 ± 0.16	Docosanoic acid	2,501	
2.13 ± 1.66	0.55 ± 0.37	Tetracosanoic acid	2,536	
31.14 ± 13.14	29.67 ± 31.81	Cholesterol	3,133	
4.73 ± 4.46	0.11 ± 0.19	Cholestanol	3,143	
1.11 ± 1.39	1.32 ± 0.75	Cholestanone	3,187	
–	4.11 ± 5.3	Hexadecyl hexadecanoate	3,361	
Note:

Retention indices have been calculated using a method developed by Van den Dool & Kratz (1963). Compounds specific for land iguanas have been given in bold (aldehydes are marked with an asterisk).

Morphological dataset

Morphological data and femoral secretions were collected in a group of territorial males to explore whether the chemical compounds might inform of certain morphological traits relevant for mate choice and/or dominance status. For this part, we focused on a single population (i.e. Amblyrhynchus cristatus mertensi from La Lobería, San Cristóbal Island).

During December 2014 and January 2015, a total of 53 male marine iguanas were measured for their snout-ventral length (SVL) using a metric-tape (±1 mm), and for head width (distance across the jaw at the widest point) using a calliper (±0.1 mm). Body mass was recorded using a portable field scale (±10 g). In addition, femoral gland secretions were also taken from all captured iguanas.

We estimated the relative head size as the residuals of a linear regression of log-transformed head size versus SVL. The body condition was calculated as the residual from a linear regression between body mass and SVL (both log-transformed) (Krebs & Singleton, 1993; Wikelski & Romero, 2003).

Behavioural dataset

In order to understand the role of chemical signals in marine iguana lek-systems, we performed an observational study at the La Lobería colony in San Cristóbal during the period from 14/12/2015 until 06/01/2016.

The study consisted of two parts performed in parallel: (1) a count to estimate the number of iguanas in the study area was conducted; (2) the behaviour of territorial males in selected leks was recorded and their femoral glands were sampled.

Marine iguana count

A daily count was performed to provide a rough overview of the number of marine iguanas in our study site (in total 12 days). In each count, the observer walked along the coastline, counting all iguanas visible within the range of a ∼1.5 km transect. The marine iguanas were classified in three categories on the basis of their external features: territorial males, female-sized individuals and juveniles, following the classification characteristics of MacLeod et al. (2016). Territorial males have enlarged dorsal crests, large body size and conspicuous breeding coloration. Female-sized individuals lack such features, with their mostly dark and uniform coloration that contrasts with the colourful red and green patches of the males. However, female-mimicking males or sneakers cannot be identified by external features (Wikelski, Carbone & Trillmich, 1996) and therefore we were not able to distinguish sneaker males from females and both were classified as “female-sized” individuals. Juveniles can be easily distinguished by their smaller size (approximately less than 70 cm in total length). Average numbers and standard deviations (SD) are given for each category in the results section. The marine iguana count performed in this study intended to provide a general idea of the number of iguanas in our study area but population size for Amblyrhynchus cristatus in this island has been already estimated from a mark-recapture study (see MacLeod et al., 2016).

Monitoring of territorial males

The behaviour of nine territorial males was monitored in their lek-mating areas. For this, we selected two different sites at the “La Lobería” breeding colony on San Cristóbal (subspecies Amblyrhynchus cristatus mertensi) and an additional site on the neighbouring island of Santa Cruz (Amblyrhynchus cristatus hassi). Territorial males from each site were photographed at the beginning to allow identification from digital images, and marked with numbers on the flanks using non-permanent white paint to allow individual recognition during the observations. Territorial males show strong site-fidelity and occupy the same territorial areas during the entire reproductive season (Wikelski, Carbone & Trillmich, 1996). We made an overall of 55 observations on focal territorial males, each lasting between 15 and 60 min, during which we noted the number of head bobs in each observation period. Observations were made between 8 am and 6 pm local time always on sunny days. The head-bobbing rate was then calculated by dividing the total number of head bobs recorded by the total time of the observation in minutes. We also monitored the presence or absence of females in the male territory (females were considered as “present” when one or more females were within approximately two body lengths of the focal male).

Statistical Analysis

The proportions of the compounds were re-standardised by calculating the relative contribution of each compound (peak area of a focal compound) with respect to the total peak area for all substances. Statistical analyses were performed with the interface Rstudio in R software version 3.3.2 (R Development Core Team, 2013), as well as with Statistica v8.0 (Statsoft Inc., Tulsa, OK, USA).

A permutational multivariate ANOVA (PERMANOVA) was used to test for significant differences in chemical profiles among the different islands. Multiple pairwise comparisons were performed by correcting for multiple testing on the basis of the Benjamini–Hochberg (B–H) method, which controls for false-discovery rate (Benjamini & Hochberg, 1995). Additionally, we plotted the data in two dimensions using non-metric multidimensional scaling (NMDS). We used SIMPER (similarity percentage analysis) to assess which compounds contributed most to differences among populations (package “vegan”; Oksanen et al., 2011). This analysis allows identifying which compounds contribute more to the observed pattern of similarity.

Furthermore, the relationship between morphological and behavioural variables, and the proportion of specific lipids was examined in marine iguanas. In order to explore the degree of collinearity in chemical composition data, a principal component analysis (PCA) was performed on the 13 lipophilic compounds, including those from the individuals for which morphological or behavioural data had been gathered. We used the PCA to reduce the dimensionality of the data, and to detect co-linearity among the levels of the various compounds. Therefore, to avoid including co-linearity among independent variables in subsequent model development, only those compounds that correlated best with each PC factor (i.e. variables with the highest loadings) were used, and the remaining compounds were discarded (following (Dormann et al., 2013); see Table S2). Compounds that correlated strongly with each PC factor were selected as independent variables for further modelling, these were: tetracosanoic acid, hexadecanoic acid, 11-eicosenoic acid and 9-octadecenoic acid.

In order to test the relationship between morphology and proportion of the volatile compounds in secretions, we built three linear models which considered body size (SVL), body condition, and relative head size as dependent variables, with the proportion of selected compounds as explanatory variables. The probability that the chemical profile of a given male influenced whether or not it was accompanied by females was modelled using a mixed effects logistic regression (binomial family, “glmer” function of package: “lme4”; (Bates et al., 2014)). The identity of the male was considered as a random factor. The best model, including the best subset of independent variables, was selected following the AIC corrected for small sample size with the use of the “glmulti” package (Calcagno, 2013). As some iguanas had no 11-eicosenoic acid, we tested whether the presence of this chemical explained their head-bobbing behaviour by performing a non-parametric Mann–Whitney U-test.

Results

Diversity of lipids in femoral glands of Galápagos iguanas

A total of 13 lipids from marine iguanas were identified and quantified (Table 1 and Fig. 2). We characterised 10 carboxylic acids with chain length ranging between C16 and C24, as well as three steroids. The most common compound in the femoral glands was cholesterol (31.14%) followed by hexadecanoic acid (22%). C20-unsaturated fatty acids, such as 11-eicosenoic acid (0.35%) and 13-eicosenoic acid (0.1%), were the rarest compounds in marine iguana secretions.

Figure 2 Example of a chromatogram of marine iguana (Amblyrhynchus cristatus; SRL population).

Each number correspond with one of the identified compounds. 1 = hexadecanoic acid; 2 = 9-octadecenoic acid; 3= 11-octadecenoic acid; 4 = octadecanoic acid; 5 = 5,8,11,14-eicosatetraenoic acid; 6 = 11-eicosenoic acid; 7 = 13-eicosenoic acid; 8 = eicosanoic acid; 9 = docosanoic acid, 10 = tetracosanoic acid; 11 = cholesterol; 12 = cholestanol; 13 = cholestanone.

A total of 20 volatile compounds were found in the femoral glands of land iguanas. Of these compounds, eight were found in Conolophus subcristatus but not in Amblyrhynchus cristatus (Table 1). The exclusive compounds included saturated and unsaturated carboxylic acids, as well as aldehydes. Interestingly, aldehydes were totally absent in marine iguana femoral glands, and thus were specific to land iguanas. 13-Eicosenoic acid was absent in land iguanas, but present in some populations of marine iguanas (Tables 1 and 2).

Table 2 Percentage (mean ± SD) of the lipophilic compounds indentified and quantified in the distinct populations of marine iguanas (Amblyrhynchus cristatus).

	IS	MAR	PIN	GEN	SAN	ESP	FL	SFE	FDA	CRUZ	SRL	SRPC	
Hexadecanoic acid	13.17 ± 2.48	29.17 ± 15.15	38.47 ± 13.07	32.45 ± 16.63	29.68 ± 7.82	38.77 ± 9.79	11.51 ± 2.41	24.29 ± 8.72	18.91 ± 3.15	14.92 ± 2.95	15.84 ± 3.15	21.46 ± 12.49	
9-Octadecenoic acid	14.61 ± 2.53	12.48 ± 6.61	8.83 ± 8.43	12.52 ± 5.99	13.08 ± 4.54	17.53 ± 4.50	12.07 ± 1.75	15.73 ± 3.03	13.68 ± 2.27	15.16 ± 2.10	11.05 ± 6.42	15.16 ± 5.90	
11-Octadecenoic acid	2.60 ± 0.65	0.61 ± 0.95	0.00 ± 0	0.00 ± 0	1.98 ± 1.89	0.32 ± 0.90	1.94 ± 0.68	0.70 ± 1.07	2.03 ± 0.79	2.05 ± 0.36	2.59 ± 0.96	0.87 ± 0.94	
Octadecanoic acid	7.98 ± 1.42	17.42 ± 13.11	22.07 ± 13.69	16.24 ± 7.76	14.28 ± 4.04	19.80 ± 5.85	6.77 ± 1.08	10.61 ± 2.89	10.78 ± 1.88	8.32 ± 1.41	8.90 ± 1.06	12.75 ± 8.91	
5,8,11,14-Eicosatetraenoic acid	5.76 ± 1.01	4.49 ± 3.73	3.39 ± 4.55	2.10 ± 3.59	3.53 ± 2.52	4.47 ± 4.92	4.94 ± 0.81	3.28 ± 3.16	6.28 ± 1.02	6.04 ± 0.84	6.26 ± 1.21	5.23 ± 2.95	
11-Eicosenoic acid	1.00 ± 0.24	0.00 ± 0	0.00 ± 0	0.00 ± 0	0.12 ± 0.31	1.20 ± 3.39	0.39 ± 0.35	0.00 ± 0	0.08 ± 0.28	0.62 ± 0.45	0.51 ± 0.38	0.00 ± 0	
13-Eicosenoic acid	0.59 ± 0.24	0.00 ± 0	0.00 ± 0	0.00 ± 0	0.08 ± 0.22	0.00 ± 0	0.19 ± 0.27	0.00 ± 0	0.00 ± 0	0.19 ± 0.24	0.04 ± 0.13	0.00 ± 0	
Eicosanoic acid	2.75 ± 0.57	3.68 ± 3.09	2.36 ± 3.30	1.58 ± 2.91	6.56 ± 2.09	6.23 ± 5.06	2.80 ± 1.52	1.35 ± 2.04	4.12 ± 1.05	3.11 ± 0.5	3.57 ± 0.82	3.30 ± 2.11	
Docosanoic acid	1.94 ± 0.34	2.39 ± 2.75	2.14 ± 2.95	0.48 ± 1.27	3.91 ± 2.12	1.94 ± 2.90	3.12 ± 0.58	1.85 ± 2.21	2.82 ± 0.80	2.29 ± 0.33	2.71 ± 0.68	2.87 ± 1.87	
Tetracosanoic acid	1.81 ± 0.28	1.98 ± 2.27	0.87 ± 1.96	0.58 ± 1.55	4.02 ± 2.01	1.49 ± 2.80	2.76 ± 0.48	1.28 ± 1.92	2.57 ± 0.68	2.14 ± 0.36	2.91 ± 0.88	2.35 ± 1.79	
Cholesterol	38.50 ± 5.68	23.54 ± 13.93	21.93 ± 14.11	32.59 ± 16.21	18.03 ± 6.01	8.23 ± 9.22	40.41 ± 5.12	34.49 ± 6.28	33.18 ± 6.08	39.56 ± 6.61	37.65 ± 8.17	29.98 ± 16.25	
Cholestanol	7.85 ± 1.93	3.86 ± 6.30	0.00 ± 0	1.42 ± 3.7	4.56 ± 6.48	0.00 ± 0	10.53 ± 1.96	6.20 ± 5.97	4.32 ± 3.76	4.01 ± 2.08	5.45 ± 2.60	4.06 ± 2.56	
Cholestanone	1.41 ± 0.58	0.36 ± 0.89	0.00 ± 0	0.00 ± 0	0.15 ± 0.39	0.00 ± 0	2.54 ± 1.02	0.34 ± 1.03	1.21 ± 1.06	1.56 ± 0.74	2.51 ± 1.31	1.96 ± 2.41	
Notes:

Most influential compounds contributing to dissimilarity among populations (SIMPER analysis) are marked in bold.

Abbreviations for the populations: IS, Isabela; MAR, Marchena; PIN, Pinta; GEN, Genovesa; SAN, Santiago; ESP, Española; FL, Floreana; SFE, Santa Fe; FDA, Fernandina; CRUZ, Santa Cruz; SRL, San Cristóbal (La Lobería); SRPC, San Cristóbal (Punta Pitt).

Lipophilic profiles varied greatly among island populations of marine iguanas (Table 2; Fig. 3). All populations shared eight compounds, these were: Hexadecanoic acid, 9-octadecenoic acid, octadecanoic acid, 5,8,11,14-eicosatetraenoic acid, eicosanoic acid, docosanoic acid, tetracosanoic acid and cholesterol. Marine iguanas from some islands (e.g. Isabela, Santa Cruz, San Cristóbal (La Lobería), Floreana and Santiago) had the full range of lipids, thus presenting a high chemical diversity. However, other islands lacked one or more compounds (e.g. Española, Marchena, Pinta and Genovesa). This case was especially striking in Pinta and Genovesa, in which the diversity of lipids was the lowest of the entire archipelago, with only eight and nine lipophilic compounds, respectively. C20-unsaturated fatty acids (i.e. 11- and 13-eicosenoic acids) were most often absent in marine iguana populations (Table 2). Accordingly, the chemical composition was dependent on population origin (PERMANOVA, Pseudo F11,91 = 7.57, P = 0.001, Fig. 3). Pairwise comparisons revealed significant differences among lipids of some populations (Table S3). Surprisingly, secretion chemistry differed between populations inhabiting different islands, but belonging to the same genetic cluster (e.g. Fernandina-Isabela and Española-Floreana, see Tables S3 and S4). Conversely, some populations belonging to different genetic clusters had similar chemical profiles (e.g. Pinta-Santa Fe and San Cristóbal-Fernandina). In the case of the iguanas from Punta Pitt and La Lobería, which inhabit the same island (San Cristóbal) but are genetically distinct, pairwise comparisons showed that their chemical secretions are rather similar (corrected P = 0.21).

Figure 3 (A) Amount (percentage) of each chemical compound for all the sampled populations of marine iguanas (Amblyrhynchus cristatus).

Each colour represents one chemical compound. (B) Chemical composition plot, showing two-dimensional non-metric multidimensional scaling (nMDS), which is based on the Bray–Curtis similarity indices as calculated with metaMDS (in R; using the vegan package). The closer the data point, the more similar the compound. Ellipses represent CI = 95%.

Morphological and behavioural data

The body size (SVL) of focal marine iguanas ranged from 35 to 54 cm, and body mass ranged from 3 to 7.75 kg. The body condition index ranged from −0.40 (worst condition) to 0.26 (best condition). Male iguanas had a minimum head width of 3.6 cm and a maximum of 7.8 cm.

The average number of marine iguanas observed in the study area per study day was 135. Female-sized individuals were the most abundant group (average ± SD = 78 ± 8), followed by territorial males (57 ± 8) and juveniles (10 ± 4). The head-bobbing rate of the focal males ranged from 0.73 to 2.40 head bobs per minute. Some males were never accompanied by females, while others had females in their territories during all the observations (raw behavioural data is shown in Table S5).

Functional correlates of femoral secretions with behaviour

The PCA summarises the variation of the 13 chemical compounds in four PCs, accounting for a total of 75% of the variation. The first PC (eigenvalue = 3.96, variance = 30.4%) correlated best with tetracosanoic acid (coefficient loading = 0.89). The second PC (2.39, 17.23%) correlated most with hexadecanoic acid (0.89). The third PC (1.98, 15.21%) showed the highest correlation with 11-eicosenoic acid (0.89). Finally, the fourth PC (1.6, 12.3%) correlated with 9-octadecenoic acid (0.78).

The best explanatory model for male body condition included two variables: 11-eicosenoic acid and tetracosanoic acid (adjusted R2 = 0.24, F2,46 = 8.41, P < 0.001; 11-eicosenoic acid: estimate = 16.18, t = 3.64, P < 0.001; tetracosanoic acid: estimate = 5.31, t = 2.32, P = 0.02; Fig. 4). Body size (SVL) and relative head size were both independent of the proportions of the chemicals found in the femoral glands (all P > 0.10).

Figure 4 Plot showing the relation between male body condition and the amount of (A) 11-eicosenoic acid and (B) tetracosanoic acid in marine iguanas (Amblyrhynchus cristatus).

In grey 95% CI.

The best model (i.e. lowest AICc, see Table 3) that predicted the probability of having females in a male’s territory included only one of the chemical compounds, i.e. 11-eicosenoic acid. Males with higher proportions of 11-ecoisenoic acid were more likely to be accompanied by females (estimate ± SE = 496.6 ± 166.5, Z = 2.98, P = 0.003; Fig. 5).

Table 3 Top five ranked models that examined the probability that presence of females in a specific male’s territory depends on the chemical composition of its femoral secretions.

Probability of having females in male’s own territory	AICc	Delta	Weight	ER	
Variables included in the model	
11-Eicosenoic acid	36.54	0	0.36	1.00	
11-Eicosenoic acid + tetracosanoic acid	37.82	1.28	0.19	1.90	
11-Eicosenoic acid + 9-octadecenoic acid	38.36	0.54	0.14	1.31	
11-Eicosenoic acid + hexadecanoic acid	38.94	0.57	0.11	1.33	
11-Eicosenoic acid + tetracosanoic acid + hexadecanoic acid	39.93	1	0.06	1.64	
Notes:

The best model is shown in bold.

AICc, AIC corrected for small sample sizes; ER, evidence ratio.

Figure 5 Plot showing the probability of the presence of females in male territories compared to the amount of 11-eicosenoic acid in marine iguanas (Amblyrhynchus cristatus).

In grey 95% CI.

Furthermore, presence of 11-eicosenoic acid in femoral glands also correlated with an increased head-bobbing rate. Territorial males with 11-eicosenoic acid showed more head bobs per minute (median = 1.84) than those lacking that compound (median = 1) (see Table S4). However, this difference was only marginally significant (Mann–Whitney U-test: Z = −1.94, N = 7, P = 0.05, Fig. 6).

Figure 6 Box plot showing the head-bobbing rate (median ± Q1 and Q3) as a function of the presence or absence of 11-eicosenoic acid in marine iguanas (Amblyrhynchus cristatus).

Discussion

Although many researchers have focused their attention on the ecology, behaviour and microevolution of the Galápagos iguanas, studies on the chemical communication in these iconic reptiles have so far been neglected. Our study is the first to survey the lipophilic compounds of Galápagos iguanas, and to explore their possible function for intraspecific communication in Amblyrhynchus cristatus. Unlike many studies focusing on the chemistry of femoral gland secretions in lizards (reviewed in Weldon, Flachsbarth & Schulz (2008)), we surveyed not one but all major island populations of marine iguanas, providing a complete overview of the intraspecific variation of lipophilic substances in this species. In the following section, we discuss the general patterns of lipophilic diversity between land and marine iguanas, as well across marine iguana populations. Furthermore, we describe the composition of specific lipid compounds and their possible functional correlates for chemical communication in the context of lek-mating.

Diversity of femoral lipids in Galápagos iguanas

The lipid profiles of marine and land iguanas were different. A total of 13 and 20 different lipids were found in marine iguanas and land iguanas, respectively (Table 1). In detail, eight compounds (fatty acids and aldehydes) were found in land iguanas but not in marine iguanas. Most surprisingly, many of these compounds are aldehydes, a group of lipids that is completely absent in marine iguanas, making these substances a characteristic difference between land and marine iguanas. It is likely that land iguanas might have one or more enzymes that catalyse the production of aldehydes. Given that land and marine iguanas diverged around 4.5 million years ago on the Galápagos archipelago (MacLeod et al., 2015), enzymes catalysing aldehyde synthesis might have evolved independently in the land iguana lineage. Alternatively, genes for these enzymes might have been lost in the marine iguana lineage, or are now down-regulated. A transcriptome/genome analysis of both land and marine iguanas, as well as of closely related out-group taxa, such as members of the genus Ctenosaura from Central America, could shed light on these alternative explanations. Interestingly, aldehydes are present in the lizard Psammodromus algirus (Martín & López, 2006b), but not in a phylogenetically close species, Psammodromus hispanicus (López & Martín, 2009). The presence of aldehydes in Psammodromus algirus could have a functional environmental constraint, as this species inhabits grassy areas where substrate scent marking is difficult, and therefore the presence of highly volatile substances like aldehydes would be more suitable for short-range communication (Martín & López, 2014). In the case of Galápagos iguanas, ecological factors might also be possible drivers for the observed divergence in chemical signals between both species. For example, in marine iguanas, marking rocky substrates could be essential for establishing territories by dominant males. For this purpose, aldehydes may be poorly suited since they volatilise too rapidly in the high humidity conditions, while carboxylic acids and steroids would persist longer. A recent study on chemical signal functionality of terrestrial vertebrates (amniotes) found that aldehydes are emitted from the sender’s body rather than from scent marks, likely due to their susceptibility to oxidation and degradation that would limit their persistence in scent marks (Apps, Weldon & Kramer, 2015). Interestingly, aldehydes occurring in body odours are much shorter than those found in land iguanas, confirming that aldehydes are part of femoral gland secretions. However, further research should clarify their potential role in intraspecific communication. Alternatively, differences in chemical secretions between land and marine iguanas could be due to their distinct diets. Marine iguanas live in humid coastal areas, and adults dive to feed on algae (Trillmich & Trillmich, 1986). Conversely, land iguanas, are strictly terrestrial, feeding mainly on Opuntia cactus and other terrestrial vegetation (Traveset et al., 2016). Therefore, it could be that interspecific differences in secretions are related to the variation on feeding regimes between both species. Moreover, our results must be interpreted carefully because land iguanas were kept in captivity and this could affect their diet as well. Therefore, we cannot rule out that the differences on chemical secretions are plastic and a more intensive sampling in natural populations should shed light on the divergence on gland chemistry between both species.

In general terms, the femoral lipids present in Amblyrhynchus cristatus are similar to those found in other iguanids (Alberts, Pratt & Phillips, 1992), and are composed of ten carboxylic acids and three sterols. The carboxylic acid chain lengths ranged between C16 and C24. The most abundant compound was cholesterol. This sterol is also one of the most common compounds found in the femoral glands of many other lizard species (Martín & López, 2014). Cholesterol is thought to play an important role in stabilising other molecules, such as fatty acids found in gland secretions (Escobar et al., 2003; Weldon, Flachsbarth & Schulz, 2008). This would be especially important in hot habitats, such as those occupied by the marine iguanas, where high temperatures may contribute to the rapid degradation of chemical signals; therefore cholesterol could act as a protector of other compounds with potential pheromone activity. However, the chemical composition of the femoral glands varies greatly among populations of marine iguanas across the archipelago. For example, cholesterol accounted for almost 40% of the gland secretions in marine iguanas from Isabela and Santa Cruz, but was less than 20% of secretions from Pinta and Española iguanas. Other compounds, such as 11- and 13-eicosenoic acids, were absent in some of the northern islands (Marchena, Pinta, Genovesa), as well as in Santa Fe and northern San Cristóbal, but were present in low proportions in the rest of the islands. Eicosenoic acids have been reported in green iguanas but their relative amounts are much higher (1.4%) than in marine iguanas (0.4%). Such differences in secretions of both species could arise to distinct ecological or microclimatic conditions between both species, however, more research is needed to confirm this hypothesis. Similarly, other studies on Mediterranean lizards showed high variation among island populations within the same species. For instance, populations of the Balearic lizard Podarcis lilfordi from different islets showed strong differences in the lipid composition of their femoral secretions (Martín et al., 2013). Further, a study comparing gland chemistry of different populations of the Skyros wall lizard (Podarcis gaigeae), found that islet populations diverged considerably more than mainland populations (Runemark, Gabirot & Svensson, 2011). However, contrary to the latter study, which suggested that chemical profile variation reflects genetic differentiation between populations, the observed variation in marine iguana lipid profiles does not correlate with the underlying genetic population structure and ongoing differentiation/speciation processes. Based on analysis of microsatellite loci, most islands of the archipelago harbour a distinct genetic cluster of marine iguanas, with some exceptions where islands share a cluster (MacLeod et al., 2015). This is the case for iguanas from Española and Floreana, as well as for the populations from Isabela and Fernandina. In our study, the chemical signatures of the iguanas from these islands differed statistically. Moreover, a remarkable peculiarity of the genetic structure of marine iguanas is found in the island of San Cristóbal, where two genetically distinct subspecies are in an ongoing speciation process within the same island (MacLeod et al., 2015; Miralles et al., 2017). One of the most important mechanisms driving speciation through chemical signals is a mutation in a receptor that switches the specificity for a given pheromone compound (Leary et al., 2012). This process promotes the development of divergent pheromones, thereby establishing pre-mating barriers and precluding gene flow between populations. However, we found no evidence that the ongoing speciation process in the marine iguanas of San Cristóbal is reflected in the variation of lipophilic compounds. The femoral secretions of the two main colonies of the distinct genetic clusters, i.e. La Lobería and Punta Pitt, have similar chemical profiles. We therefore conclude that the profiles of lipophilic substances do not correlate with the genetic diversity of populations and therefore, lipophilic compounds might play only a minor role in prezygotic isolation between marine iguana populations. However, the role that different compounds have is still unknown. It could be that small amounts in specific compounds could play a role in keeping both populations apart. For example, 11- and 13-eicosenoic acids were present in very small amounts in La Lobería colony but not in Punta Pitt, and it could contribute to maintain reproductive isolation between them. Alternatively, among-population differences in femoral secretions could be related to distinct diets. It is known that dietary lipids can result in concomitant alterations of the lipid composition of important tissues (Simandle et al., 2001). Marine iguanas are specialised on feeding macrophytic algae (Trillmich & Trillmich, 1986; Wikelski, Gall & Trillmich, 1993) and it could possible that the different islands vary in the amount of grazeable algae leading to the observed difference in chemical composition of femoral glands.

Behavioural and functional correlates of lipophilic compounds

We found that marine iguanas with better body condition had higher amounts of 11-eicosenoic acid and tetracosanoic acid (see Fig. 4). Saturated and unsaturated fatty acids, such as 11-eicosenoic acid and tetracosanoic acid, are part of the energetic dietary lipids and fat stores of lizards (Geiser & Learmonth, 1994; Simandle et al., 2001). Therefore, femoral secretions rich in 11-eicosenoic acid and tetracosanoic acid might reflect male fat reserves, suggesting that only marine iguanas in an optimal body condition can afford to allocate these lipids to their femoral glands. Further support for this hypothesis comes from the fact that the presence of females in leks correlates with levels of 11-eicosenoic acid in the territorial male’s secretions. Male iguanas that had 11-eicosenoic acid in their secretions had a higher head-bobbing rate when compared to those that lacked this compound, though the effect was only marginal. Female marine iguanas choose territorial males for mating on the basis of their head-bobbing behaviour—the greater the investment in the display, the higher the reproductive success of the males (Vitousek et al., 2008). Thus, dominant males may also enhance their mating probability by releasing high amounts of 11-eicosenoic acid within their territories. Evidence to support this idea is provided by other reptile studies which show that unsaturated fatty acids may function as sexual attractors. For instance, male rock lizards Iberolacerta cyreni produce scent marks with more oleic acid to attract females and thereby increase their mating opportunities (Martín & López, 2010). A similar mechanism is likely to occur in the lek-mating system of marine iguanas, where scent marking would be essential in order to establish male territories (Partecke, von Haeseler & Wikelski, 2002; Wikelski, Carbone & Trillmich, 1996). Therefore, the results of this study suggest the presence of an underlying mechanism, whereby scent marks are produced by males to increase their mating opportunities and therefore their reproductive success. In order to attract females to their own territories, males would provide reliable information on their condition by the allocation of more 11-eicosenoic acid in their femoral gland secretions. However, other compounds such as docosanoic acid and 13-eicosenoic acid were strongly related to tetracosanoic acid and 11-eicosenoic acid, respectively, and thus it could be that these compounds are also involved in intraspecific communication. Therefore, our results must be carefully interpreted because other compounds are also good pheromone candidates. Docosanoic acid and especially 13-eicosenoic acid are compounds that should be investigated in future studies to test whether they play role in marine iguana chemical communication. Moreover, further experimental research is needed to understand how female marine iguanas use the information contained in male gland secretions. An experiment testing female response to different concentrations of 11-eicosenoic acids could elucidate whether such compound is a pheromone involved in mate choice.

Conclusion

Evolutionary divergence between land and marine iguanas is reflected by a remarkable divergence of lipophilic substances that might have altered in response to environmental constraints and the different life histories of both species. Differences in composition of lipophilic substances seem not to be of central importance for the maintenance of genetic population structure, or for the ongoing differentiation and speciation processes in marine iguanas. However, males might use certain compounds, such as fatty acids like 11-eicosenoic and tetracosanoic acids to signal a good body condition and to successfully attract females for lek-mating. The composition and type of lipophilic substances in the femoral secretions of marine iguanas seem to be more influenced by habitat conditions than by genetic differentiation.

Supplemental Information

Supplemental Information 1 Raw dataset of the chemical composition.

Click here for additional data file.

Supplemental Information 2 Sampled populations of A. cristatus and their coordinates.

Click here for additional data file.

Supplemental Information 3 Results of the principal component analysis (PCA), showing the factor loadings of each principal component (PC) for the lipophilic compounds.

Click here for additional data file.

Supplemental Information 4 PERMANOVA pairwise comparisons of lipid profiles across populations.

Click here for additional data file.

Supplemental Information 5 Descriptive statistics showing male behavioural data and chemical profile. Chemical distance matrix. Mantel correlation between chemical distance and genetic distance.

Click here for additional data file.

Supplemental Information 6 Descriptive statistics showing male behavioural data and chemical profile.

Click here for additional data file.

We want to thank Marcus Krüger, M. Dolores Astudillo and Rocío Ruiz for their support in the field. We are grateful for the constructive comments provided by three referees. This publication is contribution number 2173 of the Charles Darwin Foundation for the Galápagos Islands.

Additional Information and Declarations

Competing Interests

Author Contributions

Animal Ethics

Data Availability

The authors declare that they have no competing interests.

Alejandro Ibáñez conceived and designed the experiments, performed the experiments, analysed the data, wrote the paper, prepared figures and/or tables and reviewed drafts of the paper.

Markus Menke analysed the data, prepared figures and/or tables and reviewed drafts of the paper.

Galo Quezada reviewed drafts of the paper, permit issues and logistic support in the field.

Gustavo Jiménez-Uzcátegui reviewed drafts of the paper, logistic support in the field.

Stefan Schulz analysed the data, contributed reagents/materials/analysis tools, prepared figures and/or tables and reviewed drafts of the paper.

Sebastian Steinfartz conceived and designed the experiments, performed the experiments, contributed reagents/materials/analysis tools, wrote the paper and reviewed drafts of the paper.

The following information was supplied relating to ethical approvals (i.e. approving body and any reference numbers):

The Galápagos National Park authority granted the research permission for this study.

The following information was supplied regarding data availability:

The raw data has been supplied as Supplemental Dataset Files.

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
