# Peer review of "Diversity of compounds in femoral secretions of Galápagos iguanas (genera: Amblyrhynchus and Conolophus), and their potential role in sexual communication in lek-mating marine iguanas (Amblyrhynchus cristatus)"

_PeerJ, doi:10.7717/peerj.3689_

## Round 0.1 · original submission · Minor Revisions

All reviewers like your ms, and consider it an important contribution to the field. There are some useful suggestions in how to pitch your ms more generally, although this is not required for PeerJ it is a good idea.

All reviewers provide useful feedback. Of particular note is that two of them suggest that you have failed to provide sufficient discussion on the possible adaptive explanation of your results. I hope that you'll find their comments on your ms are thought provoking in how you might improve your work.

Reviewer 1 ·

Basic reporting

Fine, see remarks below.

Experimental design

Fine, but see remarks below.

Validity of the findings

Fine, but see remarks below.

Additional comments

General comments
1. This is a sound study of the chemical composition of femoral secretions in two iguanas (Amblyrhynchus cristatus and Conolophus subcristatus). The study describes variation in chemical composition among A. cristatus populations of different islands. In addition, inter-individual differences in composition are correlated to morphological and behavioural variation.
2. Other studies have tackled similar questions in other species of lizards, but mostly in other families (especially lacertids). It is good to see that iguanid species are now also being studied in this respect. This re-establishes an old tradition.
3. The introduction centres quite heavily on the marine iguana, while the broader research questions are somewhat neglected. If the purpose of the study is to deepen our understanding of the natural history of this (admittedly, iconic) species, then I think the paper should be submitted to a herpetological journal. If the idea is to test general hypotheses, using this species as a model, then you should frame the questions better, possibly formulating hypotheses.
4. The methods used and analyses performed seem sound to me. I think the idea that among populations, chemical distance and genetic distance should correlate, could be tested more strongly.
5. In the discussion, there is a slight tendency towards slipping into adaptive story telling. I think the authors should mention the possibility that the differences in composition of the secretions among islands (and individuals) are the result of plasticity (e.g. differences in food intake) and may therefore be ephemeral and not functional.
6. Finally, the authors have chosen to circumvent problems of collinearity in their data by retaining only one component (e.g. tetracosonoic acid) for further analyses. While I accept that this may be statistically correct, I think they should warn the reader that other components explain a similar amount of variation. The non-prudent reader could now jump to the conclusion that tetracosonoic acid is by far thé best candidate pheromone.
Detailed comments
1. Title : put the Latin names of the genera and species in italic.
2. Line 29. ‘hold the best promise’: this sounds as if nobody has studied squamates in this respect before.
3. Line 34. I suggest putting ‘e.g.’ before these references, as these are just a few examples of the many studies available on this topic.
4. Line 39. I think ‘Mantellid’ should not be capitalised.
5. Lines 39-41. This sentence is not very relevant. I recommend deleting it.
6. Line 42. I think that grammatically, ‘these’ refers to ‘femoral glands’ here, while it is meant to refer to ‘waxy secretions’.
7. Line 42. ‘poorly volatile’ sounds a bit awkward. Change to ‘heavy’?
8. Line 56. Taxonomists have changed the genus name of Lacerta monticola to ‘Iberolacerta’ recently.
9. Lines 77-79. Exactly how does ‘territorial settling’ connect to ‘lek-mating’? These seem to radically different behavioural strategies, so I do not understand why you mention it here.
10. Lines 96-97. Please provide a reference on the dimorphic nature of femoral glands in marine iguanas.
11. Line 108. Do you mean to say ‘femoral glands’ here, or should this be ‘femoral gland secretions’?
12. Lines 108-109. I would like to see some justification for this research goal. Why do you want to study the interpopulational variation in the composition of the secretion? What do you expect to find?
13. Line 116. ‘collected’: change to ‘performed’ or ‘carried out’?
14. Line 118. ‘analyzed together’: do you mean that you have correlated the chemical and behavioural data? Please be more specific about this.
15. Line 130. Perhaps this title should be more neutral – you do not really know (yet) that the secretions carry pheromones.
16. Lines 132-133. This period encompasses a whole calendar year, so the remark that it ‘coincides with the mating system’ sounds strange.
17. Line 145. I think that ‘these’ grammatically refers to ‘glands’ in the previous sentence, while you mean the secretions.
18. Lines 147-148. Please describe in more detail how (e.g. at what temperatures) you stored the secretions. Now you only mention the conditions during field work.
19. Line 192. How does this figure (53) relate to that on line 140 (62)?
20. Line 198. ‘residuals’ should be ‘residual’, I think (since it is ‘condition’, singular).
21. Lines 207-208. ‘to estimate the rough number’ sounds a bit awkward. Perhaps ‘to roughly estimate the number’ is better, or just drop the ‘rough’ part.
22. Line 223. I think you can safely drop the ‘relatively’ here, since ‘smaller’ is already comparative.
23. Line 241. 8 am to 6 pm is a long period. How did you control for the effect of temperature (or other sources of diurnal variation) on behaviour?
24. Line 258. Please provide a reference for the SIMPER procedure.
25. Line 291. Write ‘3’ in full here?
26. Lines 300-318. Where exactly do you use the SIMPER method mentioned in M&M?
27. Line 312. ‘between’ should be ‘among’ here.
28. Line 312. I think the introduction should explain why this is ‘surprising’.
29. Line 333. Make this simply ‘Behavioural correlates of femoral secretions’?
30. Line 337. I understand what you mean, but actually, the first PC correlated with the relative concentration of tetracosanoic acid. But I realise this makes the text somewhat lengthy.
31. Lines 335-340. I recommend including a table with the loadings of all original variables (lipophilic compound %) and the four PC axes considered. This way, the reader can judge whether the procedure used in subsequent analyses (i.e., singling out particular compounds on the basis of their factor loadings) is reliable.
32. Lines 362-365. This paragraph seems to suggest that previous studies on marine iguanas exist, but did not sample all major populations. If I understand well, there were no previous studies on A. cristatus, and the current remark (a bit unfairly) pertains to chemical communication studies in general.
33. Lines 365-368. These lines can be deleted.
34. Line 372. ‘significantly’ suggests that you statistically tested this, but this is not apparent from the Results section.
35. Lines 372-405. The explanations offered for the difference in secrete composition between marine and land iguanas seem to assume that they are (1) genetic and (2) adaptive. I think you should consider the possibility that they reflect differences in diet, that may be entirely plastic and ephemeral.
36. Lines 426-434. If data on the genetic relatedness of the populations are available, why did you not perform a thorough comparison between the chemical and genetic matrices?
37. Line 437. ‘though’ should be ‘through’
38. Lines 445-446. This conclusion may be premature. You do not have much information on the role that the respective compounds play (in isolation or in concert). Small differences in the concentration of a single compound may be important in this respect, e.g. if this compound is the actual pheromone. However, your current analysis may not have picked up such subtle differences, because other compounds (with much higher volume%) may have swamped them.
39. Lines 480-481. This is not really a conclusion. Also, the sentence is more suited for a paper in a herpetological journal.
40. Line 637. Table 1. Please provide the units of measurement
41. Lines 645-652. Table 2. It is not clear from the Results section how you arrived at the conclusion that the compounds in bold are contributing most to the among-population variation in compound composition.
42. Fig. 2. I think the legend to this figure should be expanded, especially because this analysis is not described in detail in the RES section.
43. Fig. 5. I think you could omit this graph. Also, there seems to be only one measure of dispersion, while the legend suggests there should be two?
44. Lines 722-731. I discovered this table after having read the entire manuscript. Please refer to it in the RES section (Lines 333-354). The factor loadings of docosanoic acid (-0.87), cholestenol (-0.78) and cholestanone (-0.73) on PC1 are comparable to that of tetracosanoic acid, so the emphasis on the latter in the paper seems somewhat unjustified. I would notify the reader that the focus on tetracosanoic acid is an artefact of the way you analysed your data (work with one compound in glm, rather than using the PC itself). The same can be said for components important to axis 2 and 3.
45. Lines 733-738. I wonder whether you could not present this in a more compact way (a matrix?)

·

Basic reporting

Reviewer’s report;

Diversity of compounds in femoral secretions of Galapagos iguanas (genera:
Amblyrhynchus and Conolophus), and their role in sexual communication in lek-mating marine iguanas (Amblyrhynchus cristatus)

The manuscript is discussing the femoral secretion composition of marine iguana in comparison with land iguana. The topic is interesting. Role of femoral secretion is studied and discussed in details. Samples of marine iguana are enough and collection from different islands, discussing the interference of genetic content is valuable. The work is well written and showed well designed methodology. Results are clear and discussion is well presented. I think good English is used throughout, literature references are well used, figures and tables are suitable, and results are relevant to hypothesis. I think the work can be published after considering the following minor changes.

Title;
I suggest writing the two species studied directly instead of ‘(genera: Amblyrhynchus and Conolophus)’.

Abstract;
There is no comment on the ecological implication of differences between secretions of both land and aquatic iguanas. I suggest a sentence in the abstract to focus on this.
L4; please change ‘organs’ to ‘structures’
L14; space between ‘ionization’ and ‘detector’
L16; change ‘sampled’ to ‘collected’
L30; Eicosenoic acid

Introduction;
L38; delete males, as the glands sometimes are found in both sexes.

Methods;
Number of land iguana samples collected is 3, which is very few!!!

L217; ….of MacLeod (2016).

Results
This section is well presented.

Discussion
This section presented the data very well, no comment on it.

Experimental design

The experimental design is well presented and shows original research. Research questions are well defined; Role of femoral secretion is studied and discussed in details, differences between marine and land iguanas are illustrated. The authors used GC-MS to investigate the lipophilic composition of femoral secretions which is the most appropriate instrument. Authors described methods in details.

Validity of the findings

Authors found differences between iguanas regarding lipophilic compounds and they found specific compounds for each group which supports the role played by femoral secretion in communication. The data is well presented statistically. Discussion ended with well stated conclusion. They were very clear about findings.

Additional comments

Reviewer’s report;

Diversity of compounds in femoral secretions of Galapagos iguanas (genera:
Amblyrhynchus and Conolophus), and their role in sexual communication in lek-mating marine iguanas (Amblyrhynchus cristatus)

The manuscript is discussing the femoral secretion composition of marine iguana in comparison with land iguana. The topic is interesting. Role of femoral secretion is studied and discussed in details. Samples of marine iguana are enough and collection from different islands, discussing the interference of genetic content is valuable. The work is well written and showed well designed methodology. Results are clear and discussion is well presented. I think good English is used throughout, literature references are well used, figures and tables are suitable, and results are relevant to hypothesis. I think the work can be published after considering the following minor changes.

Title;
I suggest writing the two species studied directly instead of ‘(genera: Amblyrhynchus and Conolophus)’.

Abstract;
There is no comment on the ecological implication of differences between secretions of both land and aquatic iguanas. I suggest a sentence in the abstract to focus on this.
L4; please change ‘organs’ to ‘structures’
L14; space between ‘ionization’ and ‘detector’
L16; change ‘sampled’ to ‘collected’
L30; Eicosenoic acid

Introduction;
L38; delete males, as the glands sometimes are found in both sexes.

Methods;
Number of land iguana samples collected is 3, which is very few!!!

L217; ….of MacLeod (2016).

Results
This section is well presented.

Discussion
This section presented the data very well, no comment on it.

·

Basic reporting

In the study submitted by Ibañez et al., they analyzed the composition of the chemical secretions from femoral glands for two species of iguanids (Amblyrhynchus cristatus and Conolophus subcristatus). In addition, they correlate the proportion of the present chemicals with morphological and behavioural data with the aim of disentangling the potential role of these compounds in sexual interactions. They found differences among studied populations of A. cristatus, and between overall profiles of A. cristatus and C. subscristatus. Interestingly, they do not find significant differences in two subspecies inhabiting the same island, suggesting that microecological conditions can be a more crucial driver of chemical signaling differences than genetic relationships. Authors also find that those individuals with apparent better conditions to mate have high proportions of 11-eicosenoid acid in their secretions.

Some points to be considered:

- In general terms, the ms is well presented and easy to follow. However, I do recommend to put some more energy in the text itself (English, grammer, constructing of sentences, etc.), as the manuscript was sometimes not pleasant to read.

- I find several places throughout the manuscript where terms referred to chemical ecology result confusing. Authors repeatedly use different words to define the same. It must be considered that, currently, there is available bibliography to consider chemical cues, chemical signals, chemosignals and pheromones as different items. Therefore, I suggest a deep review of these terms used in the manuscript.

- Overall, figures are well presented. However, an additional chromatogram should be useful to provide readers a general idea of compounds present in secretions.
You can see some comments attached in the manuscript (optional).

Experimental design

I consider the study as relevant given that it provides information not only of two more species of lizard (which is already relevant taking into account the few species whose chemical profiles are kwon to date), but also a wide comparison among many population of the same species. I congratulate the authors of this work for the magnitude of the fieldwork, since this type of effort is often overlooked in publications.

Chemical analyses seem to be well performed but, however, I cannot understand the use of a standard compound because finally, it is not used. The use of a standard peak is not bound to affect the identification of other compounds. Anyway, authors must explain why they used it in the chemical analyses and if this standar compound was not finally used, delete the explanation in this respect in M&M.

Finally, why did not authors consider females in the comparison? authors say that females own not well developed glands. I consider very intersting to know the basis of this statement. This issue is very interesting since very often males are used as model in chemical ecology studies due to females have smaller pores than males. However, and in my experience, the size of the pores or even the colour of the secretions are not significant of being/having non-functional secretions.

Validity of the findings

The novelty of this study relies on the magnitude of the comparison among populations. I consider that experimental design is well drawn in order to investigate population differences of chemical profiles but, however, I think that the comparison with C. subscristatus is overestimated. Only three individuals, and no from natural conditions, were analyzed. I believe that the comparison between marine vs. land iguanas is interesting but, there is recommended to the authors specify that, given that information comes from only three non-natural individuals, the conclusion must be treated cautiously. Diet is also important in this point. Natural population use to have different diet than lizards inhabiting enclosure, how this could affect chemical profiles of lizards?

Finally, authors compare the proportions of some particular compounds with other already studied species. This is very interesting and necessary. However, I miss some discussion about why the species here studied have a low amount of chemicals in comparison with other lizard species.

Additional comments

This study provides interesting data on variation of the chemical composition of the femoral glands of Amblyrhynchus cristatus, which can constitute a nice contribution in the field of lizards' chemical communication.

---

## Round 0.2 · accepted · Accept

Thank you for your thorough revision with which I am happy.

A couple of minor points:

L71: are very suitable models = are suitable models (no need for the hyperbole)
L171: occupy totally distinct habitats = occupy distinct habitats
Table 1: Include species names in legend or table.
Table 2: include full genus name
All Figs: Include species name in legend